# Evaluating the feasibility, effectiveness and costs of implementing person-centred follow-up care for childhood cancer survivors in four European countries: the PanCareFollowUp Care prospective cohort study protocol

Rebecca J van Kalsbeek [1] Joke C Korevaar [2] Mieke Rijken [2,3]
Riccardo Haupt [4] Monica Muraca [4] Tomáš Kepák,[5] Katerina Kepakova,[5]
Anne Blondeel,[6] Stefan Boes [7] Line E Frederiksen [8] Samira Essiaf,[6]
Jeanette F Winther [8,9] Rosella P M G Hermens [10] Anita Kienesberger,[11]
Jacqueline J Loonen [12] Gisela Michel [7] Renée L Mulder [1]
Kylie B O'Brien [13] Helena J H van der Pal [1,14] Saskia M F Pluijm [1]
Katharina Roser [7] Roderick Skinner [15,16] Marleen Renard,[17]
Anne Uyttebroeck [17] Cecilia Follin,[18] Lars Hjorth [19]
Leontien C M Kremer [1,20]

**Correspondence to**
Rebecca J van Kalsbeek;
R.J.vanKalsbeek@prinsesmaximacentrum.nl

## ABSTRACT

**Introduction** Long-term survival after childhood cancer often comes at the expense of late, adverse health conditions. However, survivorship care is frequently not available for adult survivors in Europe. The PanCareFollowUp Consortium therefore developed the PanCareFollowUp Care Intervention, an innovative person-centred survivorship care model based on experiences in the Netherlands. This paper describes the protocol of the prospective cohort study (Care Study) to evaluate the feasibility and the health economic, clinical and patient-reported outcomes of implementing PanCareFollowUp Care as usual care in four European countries.

**Methods and analysis** In this prospective, longitudinal cohort study with at least 6 months of follow-up, 800 childhood cancer survivors will receive the PanCareFollowUp Care Intervention across four study sites in Belgium, Czech Republic, Italy and Sweden, representing different healthcare systems. The PanCareFollowUp Care Intervention will be evaluated according to the Reach, Effectiveness, Adoption, Implementation and Maintenance framework. Clinical and research data are collected through questionnaires, a clinic visit for multiple medical assessments and a follow-up call. The primary outcome is empowerment, assessed with the Health Education Impact Questionnaire. A central data centre will perform quality checks, data cleaning and data validation, and provide support in data analysis. Multilevel models will be used for repeated outcome measures, with subgroup analysis, for example, by study site, attained age, sex or diagnosis.

**Ethics and dissemination** This study will be conducted in accordance with the guidelines of Good Clinical Practice

## STRENGTHS AND LIMITATIONS OF THIS STUDY

⇒ The PanCareFollowUp Care Study is designed and conducted together with survivor representatives, ensuring the outcome measures are relevant for survivors and that PanCareFollowUp Care meets their needs and expectations.

⇒ We include survivors from four different European countries, representing a variety of healthcare systems across Europe; and their experiences are used to improve the PanCareFollowUp Care Intervention before free distribution of the materials in a Replication Manual.

⇒ The PanCareFollowUp Care Intervention is evaluated in a real-life setting with a minimal number of exclusion criteria.

⇒ Since the Care Study has a limited follow-up time, a model-based economic evaluation will complement the analyses.

⇒ Participants are their own controls and effects are evaluated as changes from baseline within an individual or institution.

and the Declaration of Helsinki. The study protocol has been reviewed and approved by all relevant ethics committees. The evidence and insights gained by this study will be summarised in a Replication Manual, also including the tools required to implement the PanCareFollowUp Care Intervention in other countries. This Replication Manual will become freely available through PanCare and will be disseminated through policy and press releases.

**Trial registration number** Netherlands Trial Register (NL8918; https://www.trialregister.nl/trial/8918).

## INTRODUCTION

Over the last decades, 5-year survival rates of childhood cancer in Europe have increased substantially, from 30% in the 1970s to 80% in the early 2000s.[1] Today, the European population of childhood cancer survivors, estimated at minimally 300 000, is rising by about 12 000 per year.[2] Yet, many survivors not only experience the burden of previous cancer diagnosis, but also face treatment-related late effects.[3 4] These may become apparent years or even decades after finishing therapy[5] and might have a significant adverse impact on quality of life.[6 7] Moreover, the transition from paediatric to adult healthcare settings often lacks continuity. As a result, many adults who survived childhood cancer have increased healthcare use and experience problems in participation, which generate a substantial burden for survivors and societies in general.[8–10] Early detection of new health conditions is essential as it could prevent further harm.[11] This requires lifelong survivorship care with frequent adaptations of the follow-up plan.

Currently, only one-third of European paediatric oncology clinics provide survivorship care to adult survivors of childhood cancer.[12] In 2006, an international group of paediatric oncologists, psychologists, nurses, epidemiologists, survivors and their parents agreed in the Erice statement that has recently been updated and reconfirmed[13 14] that follow-up care should be available and accessible for all survivors throughout their lifespan.

In the past decade, international evidence-based clinical practice guidelines have been developed to support early detection and treatment of (a)symptomatic late effects, including those developed by the International Late Effects of Childhood Cancer Guideline Harmonization Group (IGHG), sometimes in collaboration with the PanCareSurFup Project.[15–23] A European models of care guideline is published and guidelines for the transition from childhood to adult healthcare settings and health promotion are currently being developed.[24 25] Yet, implementation lags behind. Recently, a person-centred approach for survivorship care for adult survivors has been implemented in Nijmegen, the Netherlands.[26] All Dutch survivors of childhood cancer are invited for follow-up care by a long-term follow-up care clinic, in which multidisciplinary teams deliver person-centred care based on contemporary surveillance guidelines.[27] The first positive effects of this person-centred approach have been reported.[24 26] The next step is to validate this person-centred approach for survivorship care in other countries.

The PanCareFollowUp Consortium, established in 2018, is a unique multidisciplinary European collaboration between 14 project partners from 10 European countries, including survivors (www.pancarefollowup.eu).[28] The aim of the consortium is to improve the quality of life

---

### Box 1    The PanCareFollowUp Care Intervention

The PanCareFollowUp Care Intervention is based on a person-centred care model[26] that aims to meet the physical, psychological and social needs of (adult) survivors of childhood cancer through shared decision-making about prevention, surveillance and treatment options. The Care Intervention consists of three steps:

a. *Preparation of the clinic visit by both the survivor and the healthcare provider (HCP)*. The survivor provides information about their health, well-being, needs and preferences by completing the PanCareFollowUp Survivor Questionnaire. The HCP prepares a Treatment Summary describing the childhood cancer treatment that the survivor has received, reviews the relevant surveillance recommendations and the PanCareFollowUp Survivor Questionnaire provided by the survivor, and thereupon prepares the Standard Survivorship Care Plan.

b. *Clinic visit including tailored follow-up care*. After obtaining a medical history and performing a physical examination, the survivor and HCP jointly discuss the results of the Survivor Questionnaire, and the Standard Survivorship Care Plan. Together, they agree on a plan for diagnostic tests and potential referral if needed, based on surveillance guidelines or clinical indication. Based on these shared decisions, as well as potential test results, the HCP creates a Draft Individualised Survivorship Care Plan and provides tailored health education.

c. *Follow-up call*. The survivor and HCP discuss the test results and the preferred model of care for future follow-up care. The results of these shared decisions are incorporated in the final Individualised Survivorship Care Plan, which the survivor may share with other HCPs.

The PanCareFollowUp Care Intervention ends after co-creation and delivery of the Individualised Survivorship Care Plan. Survivors will thereafter remain under surveillance either at or under the guidance of their clinic, frequently adjusting their Individualised Survivorship Care Plan when needed.

---

for survivors of childhood, adolescent and young adult cancer by bringing evidence-based, person-centred care to clinical practice. The PanCareFollowUp Consortium has developed two interventions: (1) a person-centred and guideline-based model of survivorship care (PanCareFollowUp Care Intervention) (see box 1)[29] and (2) an eHealth lifestyle coaching model (PanCareFollowUp Lifestyle Intervention). The protocol of the first intervention is described in this paper (version 3, 21 January 2021), the protocol of the second one will be described separately. Both will be evaluated within the PanCareFollowUp Project. The consortium published a Care Intervention Manual that contains instructions and tools required for implementing the PanCareFollowUp Care Intervention. At the project end, Replication Manuals that contain the instructions and tools required for implementation of the PanCareFollowUp Interventions will be freely distributed.

The overall aim of the PanCareFollowUp Care Study is to evaluate the feasibility, effectiveness and costs of implementing PanCareFollowUp Care as usual care for adult survivors of childhood cancer in four study sites in four European countries. Four objectives have been

formulated: (1) To what extent is implementing PanCareFollowUp Care in the participating study sites feasible?; (2) What are the patient-reported experiences and outcomes, including survivor empowerment, of PanCareFollowUp Care and how do they change?; (3) What is the number and nature of pre-existing and new clinical events detected by PanCareFollowUp Care among participating survivors?; and (4) What are the short-term (6 months) and projected long-term costs per unit change of empowerment and other outcomes after implementing PanCareFollowUp Care from the perspective of survivors and healthcare providers (HCPs)?

## METHODS AND ANALYSIS
### Study population, setting and recruitment
Survivors fulfil the inclusion criteria if they are or have been: diagnosed with cancer before the age of 19 years; treated or registered at one of the four study sites; treated with chemotherapy and/or radiation therapy for childhood cancer with or without surgery; at least 5 years from primary cancer diagnosis; at least 1 year off treatment (also applying to treatment of subsequent benign or malignant neoplasms or relapse of the primary cancer); and currently at least 16 years of age.

Exclusion criteria consist of: being unable to complete the study questionnaires because of severe neurocognitive sequelae or insufficient understanding of the language used (even with help from another person); or having previously received complete follow-up care that is similar to the care as described in the PanCareFollowUp Care Intervention Manual (box 1).

This international prospective cohort study will be conducted at four study sites located in four European countries: Belgium, Czech Republic, Italy and Sweden. All sites currently provide long-term follow-up care, either within a paediatric (Belgium, Italy) or adult (Czech Republic, Sweden) oncology centre, using a set of (inter) national guidelines and protocols. Each study site aims to include 200 survivors who complete the study. With an estimated non-response and early drop-out (informed consent signed, but no actual participation in the study) of 40%–50% based on previous experience and an estimated late drop-out (at any point after completing the time point 1 (T1) questionnaire) of 5%–10% during the study, approximately 350–400 survivors will therefore be invited at each site. To assess the feasibility of this recruitment strategy, each centre screened their respective registries and estimated a total of 5944 eligible survivors.

Each study site developed a recruitment strategy within the prerequisites of this study that fits best within their own logistics (online supplemental appendix A). Selected survivors will be invited by an invitation letter, an invitation email or by phone (depending on the usual procedure at each study site), and receive an information sheet, including contact details for additional information and an informed consent form. Reasons for non-participation can be provided. One option of the preset reasons is 'not participating because the questionnaires are being provided via internet'. In this case, the study site may decide to offer the option for paper questionnaires. Survivors who give informed consent but do not respond to the first questionnaire, even after reminders, are considered early drop-outs and will be excluded from the study, as essential data about these survivors will not be available. The first participant was enrolled in February 2021, and on 1 March 2022, 456 participants were enrolled and completed the clinic visit. The estimated last inclusion is on 30 September 2022, with last data collection on 31 May 2023.

Participating survivors can withdraw from the study at any time if they wish. They are not obliged to provide a reason for withdrawal, although it will be asked and recorded if available. To assess representativeness of the final study sample, the four centres will provide aggregated data about their total eligible population of survivors including population distributions of sex, current age, age at diagnosis, type of cancer and distance to the late effects clinic. This will be compared with the distributions among the included survivors per clinic.

During recruitment and data collection, careful monitoring of enrolment, (non-)response, reasons for nonresponse and early and late drop-out will be performed by the four study sites in close collaboration with the central data centre at the Danish Cancer Society Research Centre.

### Intervention
Survivors of childhood cancer who receive PanCareFollowUp Care (ie, care in accordance with the PanCareFollowUp Care Intervention Manual and as outlined in box 1) will be followed up until 6 months after the clinic visit. The implementation of person-centred care in this project is facilitated by a narrated PowerPoint and an on-site workshop for all HCPs involved in the study. An add-on study investigating the feasibility of delivering PanCareFollowUp Care using the digital Survivorship Passport (SurPass) tool[30] will be conducted at the Italian clinic, where SurPass is already implemented.

### Primary and secondary outcomes
This study uses a variety of outcomes to answer the four research objectives (figure 1). These are measured from T1 before the clinic visit until T5 at 6 months after the clinic visit (figure 2). Outcomes are provided by survivors and HCPs through questionnaires, a clinic visit and diagnostic tests.

1. To what extent is implementing PanCareFollowUp Care in the participating study sites feasible?

Feasibility of implementation is of major importance to ensure sustainability of the PanCareFollowUp Care Intervention. Therefore, feasibility indicators measured by questionnaires among survivors and HCPs as well as an evaluation of barriers and facilitators are included to inform about the experiences of implementing PanCareFollowUp Care (figure 2). Items include, among others,

| **PROMs or PREMs: survivors** | Premature ovarian insufficiency (females) (d) | Neurocognitive problems: language | Telangiectasias of the eye |
|---|---|---|---|
| Empowerment (HEIQ)[a] (primary outcome) | Testosterone deficiency (males) (d) | Neurocognitive problems: memory | Xerophthalmia |
| Patient satisfaction (Satisfaction Qx)[b] | TSH deficiency (d) | Neurocognitive problems: motor integration | **Feasibility outcomes: survivor** |
| Shared decision-making (SDM-Q-9)[c] | *Gastro-intestinal* | Neurocognitive problems: processing speed | Received care according to SCP |
| Resilience (CD-RISC 25)[d] | Bowel obstruction | Psychological distress (q) | Success of communication |
| HRQoL (EQ-5D-5L, SF-36, ICECAP-A)[e] | Chronic enterocolitis | Stress-related mental disorder | Missing information |
| Psychological distress (BSI-18)[f] | Gastro-intestinal strictures or fistula | Suicidal ideation (q) | *Italian study site only:* Use of and satisfaction with SurPass |
| Post-traumatic stress symptoms (PCL-5)[g] | *Hepato-biliary* | Unemployment (q) |  |
| Distress (ET)[h] | Cholelithiasis | *Renal and urinary tract* | **Feasibility outcomes: HCP (per clinic)** |
| Fatigue (SQx + PROMIS Fatigue – Short Form 8a +)[i,j] | Hepatobiliary dysfunction (d) | Bladder fibrosis | No. of eligible survivors invited |
| Pain (BPI)[k] | Hepatocellular liver injury (stage 1) (d) | Dysfunctional voiding (q) | No. of participating survivors per time point |
| Lifestyle (SQx) | Iron overload (d) | Glomerular kidney dysfunction (d) | No. of non-responders |
| Social functioning (SQx) | Liver cirrhosis | Haemorrhagic cystitis | Reasons for non-response |
| **Clinical outcomes** | Liver fibrosis | Hydronephrosis | No. of drop-outs per time point |
| *Auditory* | Liver synthetic dysfunction (d) | Tubular kidney dysfunction (d) | Reasons for drop-outs per time point |
| Hearing loss (d + q) | *Immunological* | Vesicoureteral reflux | Composition of multidisciplinary team |
| Tinnitus (q) | Spleen problems (overwhelming infections) | *Reproductive* | Use of the SCP |
| *Cardiac* | *Musculoskeletal* | Impaired fertility (q) | Reasons for non-use of SCP, if applicable |
| Arrhythmia (d + q) | Craniofacial growth problems | Impaired spermatogenesis (males) (d + q) | Shared decision making (HCP perspective; SDM-Q-Doc)[c] |
| Cardiomyopathy (d) | Osteonecrosis | Low birth weight of offspring (females) (q) | Extent to which SCP of participating survivors has been implemented and reasons for deviating |
| Pericardial disease (d) | Reduced bone mineral density (d) | Miscarriage (females) (q) |  |
| Valvular heart disease (d) | Spine kyphosis | Physical sexual dysfunction (males) (q) | *Italian study site only:* no. of SurPasses delivered, recommendation brochures given and SurPasses shared with physicians, SurPass user statistics |
| *Dental* | Spine scoliosis | Premature birth of offspring (females) (q) |  |
| Dental caries | *Neurological* | *Respiratory* |  |
| Dental developmental problems | Cavernomas | Pulmonary dysfunction (d + q) | **Health economic outcomes: survivor** |
| Xerostomia (q) | Cerebrovascular accidents | *Subsequent neoplasm* | Time investment of survivor (preparation for clinic visit, travel, total time in clinic, follow-up appointments) |
| *Dermatologic* | Neurogenic bladder | Subsequent neoplasm (benign or malignant) (d + q) |  |
| Alopecia | Neurogenic bowel | *Vascular* | Time investment of relatives (travel, total time in clinic, follow-up appointments) |
| *Endocrine* | Optic chiasm neuropathy | Aneurysms |  |
| ACTH deficiency (d) | Pain (d) | Asymptomatic coronary artery disease | Travel costs of survivor and relatives |
| Amenorrhea (females) (q) | Peripheral motor neuropathy (q) | Carotid artery disease | Other extra costs for survivor and relatives |
| Central precocious puberty (d) | Peripheral sensory neuropathy (q) | Dyslipidaemia (d) | Loss of time for survivor and relatives at paid work or in education |
| Diabetes mellitus (d) | *Psychosocial and neurocognitive* | Hypertension |  |
| Failure in pubertal progression | Adjustment difficulties | *Visual* | **Health economic outcomes: HCP** |
| Growth hormone deficiency (d) | Anxiety (q) | Cataract | Time investment of HCP and other staff tasks related to clinic visit (preparation, clinic visit, tasks following clinic visit, follow-up call) |
| Hyperthyroidism (d) | Behavioural problems | Chronic painful eye |  |
| Hypothyroidism (peripheral) (d) | Fatigue (q) | Glaucoma |  |
| Impaired glucose metabolism (d) | Low educational status (q) | Keratitis | Costs for diagnostic and screening tests |
| LH/FSH deficiency (d) | Neurocognitive problems: academics | Lacrimal duct atrophy | Costs for other consumables for clinic visit |
| Obesity | Neurocognitive problems: attention | Maculopathy |  |
| Overweight | Neurocognitive problems: executive function | Papillopathy |  |
| Premature menopause (females) (d) | Neurocognitive problems: intelligence | Retinopathy |  |

**Figure 1** Overview of all patient-reported outcome measures (PROMs) and experience measures (PREMs), clinical outcomes, feasibility outcomes and health economic outcomes used in the Care Study. Outcomes that are specific for males or females are indicated as such between brackets. For the clinical outcomes, it is indicated whether they are assessed through a diagnostic test according to the guidelines (d), Survivor Questionnaire (q) or both (d+q). Other clinical outcomes are assessed through medical history and/or physical examination. ACTH, adrenocortotropic hormone; BPI, Brief Pain Inventory; BSI-18, Brief Symptom Inventory-18; CD-RISC 25, Connor-Davidson Resilience Scale (25 items); ET, Emotion Thermometer; HCP, healthcare provider; HEIQ, Health Education Impact Questionnaire; HRQoL, health-related quality of life; ICECAP-A, ICEpop CAPability measure for Adults; LH/FSH, luteinising hormone/follicle-stimulating hormone; PCL-5, Post-Traumatic Stress Disorder Checklist for the Diagnostic and Statistical Manual of Mental Disorders, Fifth Edition; PROMIS, Patient-Reported Outcomes Measurement Information System; Satisfaction Qx, Satisfaction Questionnaire by Blaauwbroek *et al*; SCP, Survivorship Care Plan; SDM-Q-9, nine-item Shared Decision-Making Questionnaire (patient perspective); SDM-Q-Doc, nine-item Shared Decision-Making Questionnaire (HCP perspective); SF-36, Short Form-36 (36 items, version 1); SQx, Survivor Questionnaire (part of the PanCareFollowUp Care Intervention); SurPass, Survivorship Passport; TSH, thyroid-stimulating hormone. References: [a]Brunet J *et al*.[35] 2015; Osborne RH *et al*.[31] 2007. [b]Blaauwbroek R *et al*.[38] 2008 . [c]Kriston L *et al*.[39] 2010; Rodenburg-Vandenbussche S *et al*.[40] 2015. [d]Connor KM *et al*.[41] 2003. [e]EQ-5D-5L: Herdman M, *et al*.[42] 2011; SF-36: Ware JE, Jr, *et al*.[43] 1998; ICECAP-A: Al-Janabi, H *et al*.[44] 2012. [f]Derogatis LR[45] 2000. [g]Blevins CA *et al*.[46] 2015. [h]Mitchell AJ, *et al*.[47] 2010, Mitchell AJ *et al*.[48] 2010. [i]Christen S *et al*.[22] 2020. [j]Bingham Iii, CO *et al*.[49] 2019. [k]Cleeland,CS *et al*.[50] 1994.

drop-outs at different time points, use of and experiences with the Survivorship Care Plan, and shared-decision making (figure 1).

2. What are the experiences and outcomes as reported by participating survivors receiving PanCareFollowUp Care?

The primary outcome for this study is empowerment measured by the Health Education Impact Questionnaire (HEIQ).[31] Empowerment has been defined by the European Union (EU) Joint Action on Patient Safety and Quality of Care as a 'multidimensional process that helps people gain control over their own lives and increase their capacity to act on issues that they themselves define as important', a definition adapted from Luttrell *et al*.[32 33] Empowerment has been selected as the primary outcome because childhood cancer survivors encounter several transition moments starting from diagnosis, after which a greater responsibility for their own health and care is required. It is essential that survivors receive the support they need to manage and advocate for their needs. Moreover, empowerment is important to manage future health problems. We have included six of the eight scales of the HEIQ relevant to cancer survivors in our study (social integration and support, health service navigation, constructive attitudes and approaches, skill and technique acquisition, emotional distress, self-monitoring and insight). The HEIQ has previously been used in cancer patient and survivor populations.[34–36] It allows to calculate a mean for each scale indicating higher or lower empowerment in the respective domain within a participant compared with the baseline assessment.

Secondary outcomes consist of a variety of patient-reported experience and outcome measures (PREMs and PROMs), such as satisfaction and quality of life (figure 1).

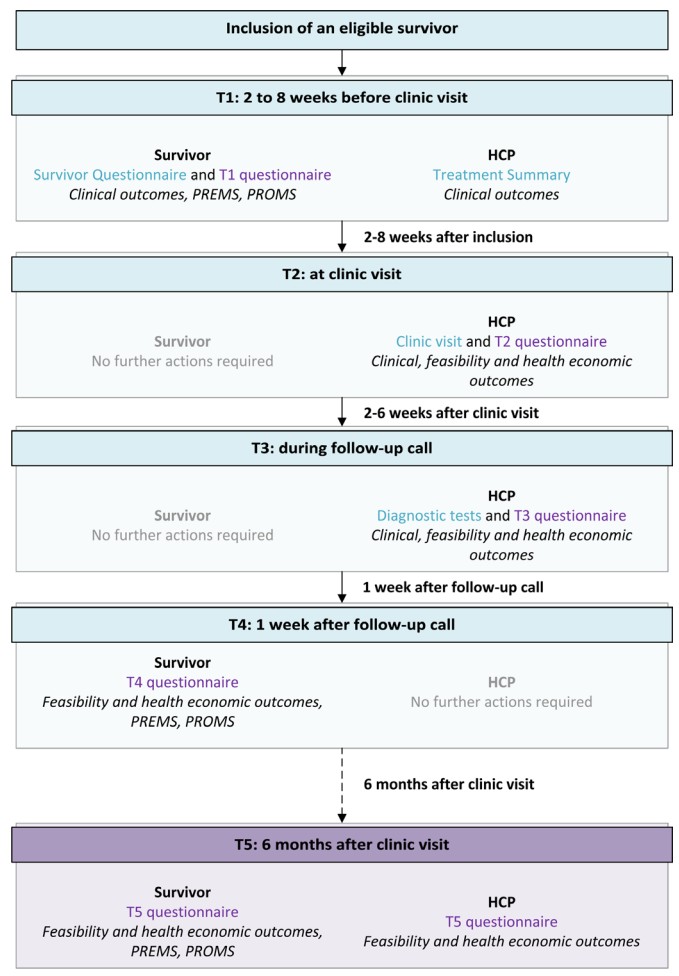

**Figure 2** Flow chart of data collection after inclusion of an eligible survivor. The boxes describe for each time point the timing of data collection, the person providing data (survivor, HCP or both), the data collection instruments (Survivor Questionnaire, Treatment Summary or T1–T5 study questionnaire) and the types of outcomes collected. Depicted in blue are data collected for care, and in purple for research purposes. HCP, healthcare provider; PREMs, patient-reported experience measures; PROMs, patient-reported outcome measures; T1, time point 1; T2, time point 2; T3, time point 3; T4, time point 4; T5, time point 5.

3. What is the number and nature of pre-existing and new clinical events detected by PanCareFollowUp Care among participating survivors?

Clinical outcomes are outcomes of symptoms and diseases and have been defined based on published or almost published guidelines of the IGHG and the PanCareFollowUp recommendations. A total of 116 clinical outcomes were defined, which reflects the wide range of late effects that survivors may encounter affecting both physical health and psychosocial well-being (figure 1). Clinical outcomes, including medical history, are collected through survivor self-report in the Survivor Questionnaire (with verification at the clinic visit), physician report in the Treatment Summary, after the clinic visit and after potential diagnostic tests (figure 2). The number and range of pre-existing and newly detected health problems (symptomatic and asymptomatic) per survivor will be described, including the results of clinical examinations (eg, echocardiogram or blood tests).

4. What are the short-term (6 months) and projected long-term costs per unit change of empowerment and other outcomes after implementing PanCareFollowUp Care from the perspective of survivors and HCPs?

The costs associated with implementing the care model will be determined by using health economic outcomes (figure 1). These reflect the time, time off work and monetary investments made by the survivor, accompanying relatives or friends, the HCP and other staff in relation to the clinic visit while receiving or providing PanCareFollowUp Care, and are collected using questionnaires (figure 2). We do not take costs outside the clinic visit into account, that is, costs related to possible primary care physician visits, mental health services or referrals to other specialists outside the clinical setting. Costs related to the clinic visit, as associated with PanCareFollowUp Care, are compared with potential benefits measured in terms of PREMs and PROMs.

An overall evaluation of implementing the PanCareFollowUp Care Intervention will be performed throughout the project according to the Reach, Effectiveness, Adoption, Implementation and Maintenance framework to assess the impact (www.re-aim.org)[37] (table 1).

### Patient and public involvement

Survivor representatives from Childhood Cancer International-Europe are included in the project as members of the PanCareFollowUp Consortium.[28] They are involved throughout the project and reach out to their respective national and international networks when needed. Survivors were involved in setting the research agenda by writing the grant application and the study protocol, developing and reviewing the PanCareFollowUp Care Intervention materials, evaluating the study questionnaires, monitoring the progress of the PanCareFollowUp Care Study and creating awareness on social media.[29] They helped consider ways to mitigate the burden of completing the study questionnaires or remembering the childhood cancer history for participants. After the end of data collection, survivor representatives will be involved in the interpretation of the study results and dissemination to participants, survivor networks and the general public.

### Power calculation

We aim to include 200 participants at each of the four study sites (total n=800). The primary outcome measure is change in empowerment between T1 and T5 as measured by the HEIQ.[34] We use six constructs (cancer version including five constructs plus one additional construct, namely self-monitoring and insight) with mean scores ranging from 2.9 (SD: 0.64) to 3.2 (SD: 0.48). Taking the construct with the largest SD (thus needing the highest number of participants to demonstrate a statistically significant change), limiting it to a single-study site, with a

**Table 1** RE-AIM framework applied to the PanCareFollowUp Care Intervention

| Components | Related outcomes/actions in the Care Study |
|---|---|
| Reach | ▶ Number and proportion of participants versus non-responders<br>▶ Representativeness of participating survivors*<br>▶ Reasons for (non-)participation |
| Effectiveness/efficacy | ▶ Main outcome empowerment*<br>▶ Patient-reported outcome and experience measures, and clinical, feasibility and health economic outcomes* |
| Adoption† | ▶ Multidisciplinarity of HCPs involved<br>▶ Recruitment rate<br>▶ Barriers and facilitators for recruitment |
| Implementation† | ▶ Use of SCP and reasons for non-use<br>▶ Adaptations made to the PanCareFollowUp Care Intervention or implementation strategy<br>▶ Time and costs of PanCareFollowUp Care for survivors and HCPs<br>▶ Barriers and facilitators for implementation |
| Maintenance | ▶ Replication Manual including updated implementation and recruitment strategy, publicly available for current and new centres<br>▶ Overview of requirements for study sites to make the PanCareFollowUp Care Intervention routine care |

*Comparisons will be made according to subgroups of sex, current age, age at diagnosis and type of cancer.
†This information will be collected at each study site separately.
HCPs, healthcare providers; RE-AIM, Reach, Effectiveness, Adoption, Implementation and Maintenance; SCP, Survivorship Care Plan.

two-sided α of 0.05 and a power of 80%, we will need 200 participants to identify an effect size of 0.2 given a mean score of 2.9 (SD: 0.64). That is enough power to demonstrate a small to medium effect. The actual power is larger since we ignored measuring empowerment repeatedly, having four centres (800 patients instead of 200) and using constructs with smaller SDs.

### Data collection

Data will be collected from participating survivors as well as from their HCPs at five time points (T1–T5) during a follow-up period of 6–8 months (figure 2). We will use data collected in the context of care delivery, and combine them with additional data collected specifically for research purposes. For the latter, there are three data collection moments for survivors and four for HCPs. These time points are linked to the structure of the PanCareFollowUp Care Intervention, which consists of three steps: (1) preparation of the clinic visit by survivor and HCP (corresponding with T1), (2) clinic visit (corresponding with T2) and (3) follow-up call (2–4 weeks after T2, corresponding with T3). Thereafter, there is data collection at 1 week after the follow-up call (T4) and 6 months after the clinic visit (T5).

The main data collection instruments consist of the PanCareFollowUp Survivor Questionnaire (care), the Treatment Summary (care), medical history, physical examinations and diagnostic tests during and after the clinic visit (care), and additional online study questionnaires for survivors and HCPs (research). The Survivor Questionnaire and Treatment Summary are available through open access.[29] The English versions of the study questionnaires for survivors have been pretested by three survivors, whereas the English questionnaires for HCPs have been pretested with at least two HCPs in each centre before the start of the data collection. The questionnaires for survivors have subsequently been translated to the local languages of the study sites, that is, Czech, Dutch, Italian and Swedish.

### Statistical analysis

For analysing outcomes measured multiple times, like the primary outcome, we will use multilevel models for repeated measures applying a fixed effect to control for study site. Next, we will perform subgroup analyses for relevant groups by including interaction terms. These subgroups will be identified based on the literature combined with knowledge from professionals. The final selection will be determined during the study. However, possible subgroups may be distinguished according to study site, sex, time since cancer diagnosis, treatment type or distance to late effects clinic. The models will be adjusted for confounders, which will be identified during the study based on the literature and expert opinion. Clinical findings will be described at each time point, like the number of prevalent conditions as well as new diseases detected, diagnoses of subclinical diseases, relapse of the original tumour, late effects and diagnostic measurements. The results will be adjusted for multiple testing.

For the health economic evaluation, we will calculate the costs associated with the implementation of the PanCareFollowUp Care Intervention in order to achieve change in different outcomes. The analysis of costs and benefits will be based on within-subject changes until 6 months of follow-up, and on model-based evaluations for longer-term predictions. The estimated benefits of the intervention are measured in terms of empowerment (HEIQ) and quality of life (Short-Form 36, EQ-5D-5L, ICEpop CAPability measure for Adults). Costs include resources incurred at the level of the hospital and the survivor. At the hospital level, we measure the time of physicians and other hospital staff for tasks related to the clinic visit and the follow-up call, costs for diagnostic and screening tests and other consumables for the clinic visit. At the survivor level, we measure the time investment and travel costs of survivors and relatives or friends, and loss of productive time at the workplace or in education. These costs are investigated separately on

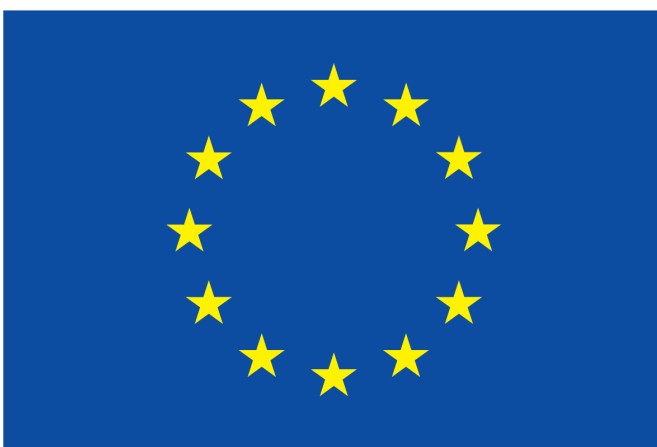

**Figure 3** European Union emblem.

each level, hospital and survivor, as well as on an aggregated level.

The calculation of cost per unit change of outcomes needs to be interpreted in light of the relatively short follow-up period of 6 months within the study. This implies that the cost evaluation mainly focuses on short-run effects, while longer-run effects of PanCare-FollowUp Care on outcomes such as survival cannot be measured within the study. Moreover, effects on other outcomes such as quality of life may be small. In order to provide information about the potential medium to long-run effects, we will complement our analysis with a model-based economic evaluation approach using data from this study as well as information from the literature on longer-term effects of follow-up interventions and patient pathways, as well as related cost estimations. This will allow us to gain a more comprehensive picture on the costs associated with the implementation of PanCareFollowUp Care.

### Handling missing data

Automated reminders and phone calls by the clinics are used to ensure that all patients and HCPs complete all questionnaires to minimise the number of missing data. In case of missing data for certain PROMs and PREMs, we will replace missing values with the mean of the remaining items of the scale as recommended by the manuals. In case of other missing data, we will perform sensitivity analyses, that is, perform the analyses with the complete cases and repeat the analyses with imputed values.

### Data management

A cloud-based Electronic Data Capture (EDC) platform has been developed by the Danish Cancer Society using Castor EDC (www.castoredc.com). This platform can be accessed by each of the four study sites for data entry. Castor EDC is compliant with all the important regulations regarding research: General Data Protection Regulation (GDPR), ISO 27001 and ISO 9001 with servers located in the Netherlands including several measures to ensure security, adequacy and veracity of the collected data: regular back-ups (four times per day); personal

accounts with individual user rights; audit, data and edit trail of all entered and changed data; and real-time edit checks to identify discrepancies in entered data.

Participating survivors complete their questionnaires directly in Castor EDC through a personalised link they receive by email. Clinical data will be provided by HCPs or retrieved from survivors' medical records and entered into Castor EDC by local data managers according to a data entry instruction manual. All personal and sensitive data collected in the PanCareFollowUp Project will be pseudonymised.

After the end of the data collection period, data will be exported from Castor to servers at the Danish Cancer Society. Experienced data managers will perform quality checks, data cleaning, and validation of data collected at the four sites and will set up data for the respective statistical analyses as subsets of the main database, governed by Data Transfer Agreements. The investigators will properly address all the ethical, legal, and safety aspects of the study and comply fully with EU Regulation 2016/679 on the protection of natural persons with regard to the processing of personal data and on the free movement of such data, and repealing Directive 95/46/EC (GDPR).

### ETHICS AND DISSEMINATION

This study will be conducted in accordance with the guidelines of Good Clinical Practice by the International Council for Harmonisation of Technical Requirements for Pharmaceuticals for Human Use and the Declaration of Helsinki, written to protect those involved in clinical studies. The study protocol has been reviewed and approved by all relevant ethics committees: Brno, Ethics Committee of St Anne's University Hospital (13 August 2019); Leuven, Ethics Committee Research University Hospitals Leuven (16 December 2020); Stockholm, Ethics Review Authority Stockholm (26 October 2020); Genoa, N Liguria Regional Ethics Committee (13 July 2020).

Written informed consent will be obtained from all study participants before enrolment and data collection. An independent ethics advisor from Denmark is available to provide feedback and advice on ethics issues that may arise. An external study steering committee has been appointed to act as an advisory capacity with study oversight and external advice. The committee includes a survivor representative, a clinical oncologist, a late effects specialist, an ethicist and a statistician.

Incidental findings based on participants' completion of the questionnaires are unlikely given the nature of the questions, except for one question of the Brief Symptom Inventory-18 on suicidal thoughts. The central data centre and the four study sites will regularly check for any positive answers on this specific question, and inform the HCP as soon as possible, but within a maximum of 2 weeks. Worrisome answers at the pre-visit questionnaire will be discussed at the clinic visit. In the post-visit questionnaires, the survivor is informed that he or she can

contact their general physician or late effects clinic in case of worrisome symptoms or complaints.

After the project, a Replication Manual will be developed for anyone interested in implementing PanCareFollowUp Care for adult survivors of childhood cancer. It will include an updated Intervention Manual based on the Care Study results and additional focus groups with project stakeholders after the study closes. The Replication Manual will include all materials required for implementation in different languages and will become freely available through PanCare. PanCareFollowUp is aligned with EC Open Science Initiative, providing open access to all publications, and participates in the H2020 Open Research Data Pilot. The PanCareFollowUp Consortium will ensure that the collected data are findable, accessible, interoperable and reusable. A dissemination plan including policy and press releases has been created warranting publications and lay language summaries on the different outcomes collected, to be distributed through the networks of PanCare and several (inter) national childhood cancer organisations. In addition, results will be published in peer-reviewed journals and presented on the project website.

## Disclaimer
The material presented and views expressed here are the responsibility of the author(s) only. The EU Commission takes no responsibility for any use made of the information set out (figure 3).

**Author affiliations**
[1]Princess Maxima Center for Pediatric Oncology, Utrecht, The Netherlands
[2]Netherlands Institute for Health Services Research (NIVEL), Utrecht, The Netherlands
[3]Department of Health and Social Care Management, University of Eastern Finland-Kuopio Campus, Kuopio, Finland
[4]DOPO Clinic, Department of Hematology/Oncology, IRCCS Istituto Giannina Gaslini, Genoa, Italy
[5]International Clinical Research Centre (FNUSA-ICRC) at St Anne's University Hospital, Masaryk University Faculty of Medicine, Brno, Czech Republic
[6]European Society for Pediatric Oncology (SIOP Europe), Brussels, Belgium
[7]Department of Health Sciences and Medicine, University of Lucerne, Lucerne, Switzerland
[8]Childhood Cancer Research Group, Danish Cancer Society Research Center, Copenhagen, Denmark
[9]Department of Clinical Medicine and Faculty of Health, Aarhus Universitet, Aarhus, Denmark
[10]Scientific Institute for Quality of Healthcare (IQ Healthcare), Radboudumc, Nijmegen, The Netherlands
[11]Childhood Cancer International Europe, Vienna, Austria
[12]Department of Hematology, Radboudumc, Nijmegen, The Netherlands
[13]Pintail, Limited, Dublin, Ireland
[14]PanCare, Bussum, The Netherlands
[15]Wolfson Childhood Cancer Research Centre, Newcastle University Centre for Cancer, Newcastle upon Tyne, UK
[16]Royal Victoria Infirmary, Great North Children's Hospital, Newcastle upon Tyne, UK
[17]Department of Paediatric Haematology and Oncology, KU Leuven, University Hospitals Leuven, Leuven, Belgium
[18]Department of Clinical Sciences Lund, Oncology, Lund University, Skane University Hospital, Lund, Sweden
[19]Department of Clinical Sciences Lund, Paediatrics, Lund University, Skane University Hospital, Lund, Sweden
[20]Department of Paediatrics, Emma Children's Hospital, Amsterdam UMC, University of Amsterdam, Amsterdam, The Netherlands

**Acknowledgements** The authors gratefully acknowledge Carina Schneider and Patricia McColgan for organising pilot tests of the study questionnaires with survivors, and Aoife Moggan, Rory McGrath and Daniel Owens for their participation in the pilot test. We acknowledge Anja Krøyer, Thomas Tjørnelund Nielsen and Agnethe Kirstine Møldrup Poulsen from the Childhood Cancer Research Group at the Danish Cancer Society Research Centre, Copenhagen, Denmark for setting up the PanCareFollowUp Care Study Castor EDC database.

**Contributors** RJvK, JCK, MR and LK contributed to the conception and design of the work and drafted and substantially revised the manuscript. RH, MM, TK, KK, AB, SB, LEF, SE, JFW, RH, AK, JL, GM, RM, KO, HvdP, SP, KR, RS, MR, AU, CF and LH contributed to the conception and design of the work and critically revised the manuscript. All authors read and approved of the final manuscript.

**Funding** This work was supported by the European Union's Horizon 2020 research and innovation programme (grant number 824982), the Swedish Childhood Cancer Fund (grant number EU 2018-0002) and the Italian Ministry of Health (grant number not applicable).

**Disclaimer** The funding bodies and primary sponsor had no role in the design of the study; in the collection, management, analysis and interpretation of data; in writing of the report; or in the decision to submit the report for publication.

**Competing interests** None declared.

**Patient and public involvement** Patients and/or the public were involved in the design, or conduct, or reporting, or dissemination plans of this research. Refer to the Methods section for further details.

**Patient consent for publication** Not required.

**Provenance and peer review** Not commissioned; externally peer reviewed.

**ORCID iDs**
Rebecca J van Kalsbeek http://orcid.org/0000-0003-3407-6508
Joke C Korevaar http://orcid.org/0000-0001-9997-040X
Mieke Rijken http://orcid.org/0000-0001-6070-4091
Riccardo Haupt http://orcid.org/0000-0003-0571-8460
Monica Muraca http://orcid.org/0000-0001-5259-7046
Stefan Boes http://orcid.org/0000-0001-5478-9105
Line E Frederiksen http://orcid.org/0000-0003-3841-9053
Jeanette F Winther http://orcid.org/0000-0002-3440-5108
Rosella P M G Hermens http://orcid.org/0000-0001-7624-7120
Jacqueline J Loonen http://orcid.org/0000-0002-9963-8367
Gisela Michel http://orcid.org/0000-0002-9589-0928
Renée L Mulder http://orcid.org/0000-0002-0414-9561
Kylie B O'Brien http://orcid.org/0000-0002-4412-1483
Helena J H van der Pal http://orcid.org/0000-0003-2253-2115
Saskia M F Pluijm http://orcid.org/0000-0002-4459-7799
Katharina Roser http://orcid.org/0000-0001-5253-3333
Roderick Skinner http://orcid.org/0000-0002-1162-675X
Anne Uyttebroeck http://orcid.org/0000-0001-5644-424X
Lars Hjorth http://orcid.org/0000-0002-8302-7174
Leontien C M Kremer http://orcid.org/0000-0001-7422-3248

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
