## [Reviewer comments · BMJ Open]

ARTICLE DETAILS

TITLE (PROVISIONAL)	Evaluating the feasibility, effectiveness and costs of implementing person-centred follow-up care for childhood cancer survivors in four European countries: the PanCareFollowUp Care prospective cohort study protocol
AUTHORS	van Kalsbeek, Rebecca; Korevaar, J; Rijken, Mieke; Haupt, Riccardo; Muraca, Monica; Kepák, Tomáš; Kepakova, Katerina; Blondeel, Anne; Boes, Stefan; Frederiksen, Line Elmerdahl; Essiaf, Samira; Winther, Jeanette; Hermens, Rosella; Kienesberger, Anita; Loonen, Jacqueline; Michel, Gisela; Mulder, Renée; O'Brien, Kylie; van der Pal, Helena; Pluijm, Saskia; Roser, Katharina; Skinner, Roderick; Renard, Marleen; Uyttebroeck, Anne; Follin, Cecilia; Hjorth, Lars; Kremer, Leontien

VERSION 1 – REVIEW

REVIEWER	Horton, Susan University of Waterloo, School of Public Health and Health Systems
REVIEW RETURNED	06-Apr-2022

GENERAL COMMENTS	Review: BMJOpen Childhood cancer follow-up Disclaimer: since I am an economist, my expertise is greater in some areas of the protocol than others. I have various concerns: The period of follow-up is very short (6 months), hence why psycho-social outcomes (empowerment) are prioritized over health outcomes used in previous studies with a longer time horizon and which have conducted cost-effectiveness evaluations. How do the authors plan to account for “decay” in response to an educational-type intervention? Evidence from other lifestyle interventions (e.g. physical exercise, eating behavior, diabetes care) suggests that benefits “decay” over time, and that cost-effectiveness estimates conducted a short time after the intervention (with optimistic projections about maintenance of impact) are over-optimistic. The group is very heterogeneous – in type of cancer, time since diagnosis, time since last treatment, intensity of treatment received etc. This may make it easier for the researchers to identify those groups most likely to benefit from such an intervention, but more difficult to ascertain the actual impact. I would imagine that the benefits of the intervention will differ depending on length of time since treatment completion – that more impact would be realized if the intervention occurred closer to that time. The researchers do plan to control for these confounders. They don't mention controlling for risk, which reference (26) notes is a really important variable in affecting outcomes and late effects. Perhaps this might be considered?
--

	I didn't really understand how the Belgium sample could fulfil the same inclusion criteria as the other countries. I thought I understood that survivors had to be at least 16, within five years of diagnosis, and at least one year past any relapse/complications. Then I read that in Belgium the first group to be invited would be those diagnosed prior to 1990, followed by those diagnosed between 1991 and 2000, and finally those diagnosed after 2000. Perhaps reviewers with clinical background would understand this better than me? I did not understand the rationale for excluding secondary costs (page 14, line 14, referring to subsequent clinical visits and diagnostic tests incurred following up from the intervention). To an economist, these are costs to the health care system and to society and should be included in the analysis from those two perspectives. It sounds as if the intervention might be very valuable and worth disseminating more broadly. I can see the value of a heterogeneous patient population in order to identify who will benefit most from this intervention. But I am skeptical of the value of the proposed cost-effectiveness estimates, which will rely on modelling, which in turn will rely on assumptions about key parameters (in particular how much "empowerment" decays over time without reinforcement). Personally I would recommend using this very interesting dataset to identify who is likely to benefit most from the intervention, and then undertake a longer-term study in a more focused population, prior to doing the economic calculations.
--	--

REVIEWER	Rodwin, Rozalyn Yale School of Medicine
REVIEW RETURNED	09-Sep-2022

GENERAL COMMENTS	Overall interesting and well written protocol. Primary comment is to add clarification within text of protocol of how and when outcomes are obtained to supplement Figure 1.  - For feasibility outcomes would be helpful to expand on the specifics outcomes assessed (how will use of/experience with survivorship care plan and use of shared decision making be quantified?) - Would also be helpful to briefly describe HEIQ questionnaire since this is the primary outcome (is it specific to cancer survivors or other populations, is there a cut-off for impairment, does it assess multiple domains) - Include in text specific measures that will be used to measure satisfaction and quality of life as secondary outcomes or reference Figure 1 - Add line in text describing how clinical outcomes will be identified (patient report, physician report, lab/imaging evals), and at what time points will they be assessed in addition to listing in Figure 1. Is it the provider at the time of the visit, patient, or someone else who reports these? And will any clinical outcomes they have ever had since cancer diagnosis be included at time-point 1 or just active clinical issues? - How will health economic outcomes be assessed? (participant survey, etc.) - Would be helpful to include description/definition of T1-T5 earlier in protocol before describing outcomes or make reference to figure earlier
---

	-Would be helpful to add SQx questionnaire as appendix if this is a new measure specific to this protocol Minor comments: -typo page 14 line 58 should be "as well as" new diseases -Acronyms like SF-36, EQ-5D-5L are first used in statistics section but should be defined
--	---

VERSION 1 – AUTHOR RESPONSE

Comments by reviewer 1

1. The period of follow-up is very short (6 months), hence why psycho-social outcomes (empowerment) are prioritized over health outcomes used in previous studies with a longer time horizon and which have conducted cost-effectiveness evaluations. How do the authors plan to account for “decay” in response to an educational-type intervention? Evidence from other lifestyle interventions (e.g. physical exercise, eating behavior, diabetes care) suggests that benefits “decay” over time, and that cost-effectiveness estimates conducted a short time after the intervention (with optimistic projections about maintenance of impact) are over-optimistic.

Response: Thank you very much for reviewing our manuscript and for providing this remark. The PanCareFollowUp Care Intervention might indeed be subject to decay in some of the outcomes, e.g. empowerment. However, we also believe that the impact of the intervention on other outcomes will remain over time, as follow-up care will continue at regular time points after the initial clinic visit, emphasizing person-centred care, i.e. follow-up that is specifically designed to overcome such decay effects.

In addition, the impact on clinical outcomes is expected to remain similar, or even increase over time. From the experience of the healthcare providers involved in the project, the difficult part is to retrace survivors lost to follow-up. Once in follow-up, survivors are very motivated to continue and to receive health surveillance for conditions they are at risk for. We know the cumulative burden of late effects increases substantially over time. By including survivors in long-term follow-up, late effects are prevented or treated in an early stage, resulting in long-term impact on their health and quality of life, especially if surveillance is continued at regular time points according to the PanCareFollowUp Survivorship Care Plan that is developed.

We agree with the point that the cost-effectiveness of the intervention can only be reported confidently over the first six months after the intervention, and thereafter we need to rely on model-based evaluations using suitable assumptions. The model-based evaluations can account for such a decay effect, e.g., as part of a scenario analysis with different forms of decay over time. The literature mentioned on lifestyle interventions could be well suited to inform the assumptions we need to make on the form of decay effect. However, we agree with the reviewer and are aware of this limitation in our study in terms of assessing long-term cost-effectiveness. For this reason, we already described this limitation in the protocol on page 15, line 22-26, including a reference to the model-based evaluations as a potential way of addressing it.

We tried to reflect the abovementioned (as well as future) remarks by the following modifications in the manuscript (page 15).

Previous version:

“In the calculation of ICERs, we will take into account the follow-up of six months, which implies that longer-run effects of PanCareFollowUp Care on outcomes such as survival cannot be measured within the study, and effects on other outcomes such as quality of life may be small. We therefore complement our analysis with a model-based economic evaluation approach using data from this study as well as information from the literature on longer-term effects of follow-up interventions and patient pathways, which will allow us to gain a more comprehensive picture on the cost-effectiveness of PanCareFollowUp Care.”

Current version (changes in orange):

“The calculation of ICERs needs to be interpreted in light of the relatively short follow-up period of six months within the study. This implies that the cost-effectiveness analysis mainly focuses on short-run effects, while longer-run effects of PanCareFollowUp Care on outcomes such as survival cannot be measured within the study. Moreover, the effects on other outcomes such as quality of life may be small. In order to provide information about the potential medium- to long-run effects, we will complement our analysis with a model-based economic evaluation approach using data from this study as well as information from the literature on longer-term effects of follow-up interventions and patient pathways, as well as related cost estimations. This will allow us to gain a more comprehensive picture on the cost-effectiveness of PanCareFollowUp Care.”

2. The group is very heterogeneous – in type of cancer, time since diagnosis, time since last treatment, intensity of treatment received etc. This may make it easier for the researchers to identify those groups most likely to benefit from such an intervention, but more difficult to ascertain the actual impact. I would imagine that the benefits of the intervention will differ depending on length of time since treatment completion – that more impact would be realized if the intervention occurred closer to that time.

Response: Thank you very much for pointing this out. The population of five-year childhood cancer survivors is indeed very heterogeneous. However, long-term follow-up care is usually organized across Europe in specialized long-term follow-up clinics that provide care for this combined group, regardless of their diagnosis. Separate studies with homogeneous diagnosis groups would not be representative for the current organizational structure, and would therefore also limit the conclusions that can be drawn about the feasibility of the implementation of a person-centred care model. In addition, late effects seem to be more strongly related to the type and intensity of treatment received than to the initial diagnosis, with risk-based surveillance guidelines mostly stratified by treatment factors.

We agree that the benefits of the intervention will probably differ depending on length of time since treatment completion, but actually expect more impact will be realized if the intervention is performed after a longer follow-up time. Survivors lost to follow-up may have a substantial burden of late effects, which is often not recognized by their regular health care providers due to lack of awareness about their childhood cancer history, or their relatively young age. The yield of surveillance and the benefits of adequate treatment can be high in this group. On the other hand, we believe that the person-centred PanCareFollowUp Care Intervention will also be cost-effective in survivors with a shorter time since diagnosis, since they will benefit most from prevention and early detection of treatable conditions, preventing worse outcomes later in life.

The abovementioned factors (type of cancer, type of treatment, time since diagnosis, time since last treatment) will be included in our analyses and important findings related to these factors will be published in our open-access publications, as well as in the Replication Manual. We will take a look at the diagnosis subgroups to see if there is an indication that some subgroups benefit more than the average.

3. The researchers do plan to control for these confounders. They don't mention controlling for risk, which reference (26) notes is a really important variable in affecting outcomes and late effects. Perhaps this might be considered?

Response: Thank you for relating reference 26 to our study protocol. The risk of late effects is mostly determined by the treatment that the survivor received for their childhood cancer. Treatment (including chemotherapy, radiotherapy, stem cell transplantation or surgery) are documented in detail by the healthcare provider as part of the Treatment Summary, and will be used in the analysis to determine their relation with outcomes and late effects. Diagnosis also has an association with the type and severity of late effects, but this is almost completely mediated through the treatment given, so treatment factors are preferred for analyses in this field of research.

4. I didn't really understand how the Belgium sample could fulfil the same inclusion criteria as the other countries. I thought I understood that survivors had to be at least 16, within five years of diagnosis, and at least one year past any relapse/complications. Then I read that in Belgium the first group to be invited would be those diagnosed prior to 1990, followed by those diagnosed between 1991 and 2000, and finally those diagnosed after 2000. Perhaps reviewers with clinical background would understand this better than me?

Response: We would like to point out that our study includes survivors after five years of diagnosis (page 9), which is the time point at which long-term follow-up is initiated. In all centres, only survivors at least five years from primary cancer diagnosis are eligible to participate.

5. I did not understand the rationale for excluding secondary costs (page 14, line 14, referring to subsequent clinical visits and diagnostic tests incurred following up from the intervention). To an economist, these are costs to the health care system and to society and should be included in the analysis from those two perspectives.

Response: Thank you for this observation. We agree that costs for subsequent clinical visits, also after referral to other specialists, and diagnostic tests contribute to the costs to the health care system and the society. However, from a feasibility perspective, it is not possible to trace participants outside of the late effects clinic. Each of the four participating centres has a different situation regarding referrals. For example, in Sweden, survivors need to be referred to a hospital that is close to their home, which may be both geographically far away and organizationally completely separated from the late effects clinic. Therefore, it is not possible to collect these costs after the late effects clinic, at least within the current project.

On the other hand, it could be argued that we also do not capture all of the benefits when measuring them through the late effects clinic only. Benefits of prevention, early detection and treatment might become apparent in 10 to 20 years' time. These limitations relate to point 1 above and the restricted time horizon of the study. In terms of the analysis, however, secondary costs due to referrals can be included in the health economic modeling based on assumptions on likely treatment paths, informed by the follow-up care visit, and information about treatment costs (at least on average) in the countries of the four participating centres. As part of the health economic model, following common practice in decision-analytic modeling, we will also conduct a sensitivity analysis to assess the extent to which our conclusions of cost-effectiveness of the follow-up care intervention depend on these secondary costs (and later benefits).

6. It sounds as if the intervention might be very valuable and worth disseminating more broadly. I can see the value of a heterogeneous patient population in order to identify who will benefit most from this intervention. But I am skeptical of the value of the proposed cost-effectiveness estimates, which will rely on modelling, which in turn will rely on assumptions about key

parameters (in particular how much “empowerment” decays over time without reinforcement). Personally I would recommend using this very interesting dataset to identify who is likely to benefit most from the intervention, and then undertake a longer-term study in a more focused population, prior to doing the economic calculations.

Response: Thank you for the interesting suggestion. The primary goal of this study is to evaluate the PanCareFollowUp Care Intervention as a novel person-centred long-term follow-up care model that is flexible to be implemented in European countries without any or complete long-term follow-up. Our main interests are whether it empower the survivor, our primary outcome, whether it is feasible to implement and meets the needs of survivors and healthcare providers. The question who specifically would benefit most from the intervention is very interesting, but was not our primary study aim. We hope that long-term follow-up care is beneficial for the entire group of childhood cancer survivors, but as suggested, we can extend our analyses to asses who benefits most. Optimal efficiency is reached through the development of evidence-based and risk-based guidelines that describe who should receive which surveillance, within a late effects clinic visit, and such a heterogeneity analysis could well contribute to the development of such guidelines. However, we also believe the PanCareFollowUp Care Intervention should be available to all childhood cancer survivors, and not just to focused populations. In terms of the economic evaluation, we agree that the short time horizon is a clear limitation, as mentioned in points 1 and 5. The goal of developing a decision-analytic model as a basis for the health economic evaluation within our project is to provide a starting point for assessing the cost-effectiveness of the PanCareFollowUp Care Intervention, which may be adapted over time and enriched with additional data that may be collected in longer-term folow-up study (see also the next point).

We have adapted the manuscript (page 15) to address these remarks more clearly, as described below point 1.

7. I believe that the protocol is suitable to assess the feasibility of assessing the intervention, but the duration will not permit an appropriate assessment of its cost-effectiveness.

Response: Thank you for sharing your conclusions. Related to our responses to the previous two points, and to point 1 above, we agree that the study duration poses a limitation to the evaluation of cost-effectiveness of the PanCareFollowUp Care Intervention, which is why cost-effectiveness is one of the secondary outcomes considered in the project. We also agree that our conclusions about cost-effectiveness need to be interpreted in the light of this limitation. Nevertheless, we consider the study of cost-effectiveness an important element of the project, especially as part of a health economic model. This model will have to account for the uncertainty we have on longer-term costs and benefits, but the model structure and initial information, for example, on patient pathways, can be inferred from the data collected during the clinic visit and within the 6-month follow-up, and the model can be updated once new information becomes available. For example, most survivors will remain in follow-up at the four clinics after this study, so future analyses can be carried out taking advantage of this data to learn more about the longer-term cost-effectiveness.

We have adapted the manuscript (page 15) to address these remarks more clearly, as described below point 1.

Comments by reviewer 2

1. Overall interesting and well written protocol. Primary comment is to add clarification within text of protocol of how and when outcomes are obtained to supplement Figure 1.

Response: Thank you very much for reviewing our manuscript and providing this suggestion for improving the manuscript. When considering your comments, we agreed that we could improve the

understanding about the outcome (which, how they were collected, by whom, and at which time points) by describing them more clearly and/or referring to the appropriate tables. For this point specifically, the measurement of the outcomes is already described in detail at a later point in the manuscript, but we agree that readers should have an overview earlier on. Therefore, we have now added two sentences in the section “Primary and secondary outcomes” (page 11).

Previous version:

“This study uses a variety of outcomes to answer the four research objectives (Figure 1).”

Current version (changes in orange):

“This study uses a variety of outcomes to answer the four research objectives). These are measured from time point 1 (T1) before the clinic visit until T5 at six months after the clinic visit (Figure 2). Outcomes are provided by survivors and HCPs through questionnaires, a clinic visit and diagnostic tests (Figure 2).”

2. For feasibility outcomes would be helpful to expand on the specifics outcomes assessed (how will use of/experience with survivorship care plan and use of shared decision making be quantified?)

Response: Thanks for noting that the exact feasibility outcomes are not clear from the manuscript itself. The feasibility outcomes measured by a questionnaire (see Figure 2: at T2, T3 and T5 for healthcare providers, and at T4 and T5 for survivors). The specific feasibility outcomes assessed are described in Figure 1, and differ for survivors (including “receiving care according to Survivorship Care Plan”, “success of communication” and “missing information”) and healthcare providers (including “no. of eligible survivors invited”, “no. of participating survivors per time point”, “no. of non-responders”, “reasons for non-response”, “no. of drop-outs per time point”, “composition of multidisciplinary team”, “use of the Survivorship Care Plan”, “reasons for non-use of Survivorship Care Plan, if applicable”, “shared decision making” measured by the SDM-Q-Doc questionnaire” and “extent to which Survivorship Care Plan of participating survivors has been implemented and reasons for deviating”).

We believe that it is important to clarify this in the text, and therefore propose the following suggestions in the manuscript (page 11).

Previous version:

“Feasibility of implementation is of major importance to ensure sustainability of the PanCareFollowUp Care Intervention. Therefore, feasibility indicators as well as an evaluation of barriers and facilitators are included to inform about the experiences of implementing PanCareFollowUp Care, both from the survivor’s and the HCP’s perspective. These include drop-outs at different time-points, use of and experiences with the Survivorship Care Plan, and shared-decision making.”

Current version (changes in orange):

“Feasibility of implementation is of major importance to ensure sustainability of the PanCareFollowUp Care Intervention. Therefore, feasibility indicators measured by questionnaires among survivors and health care providers as well as an evaluation of barriers and facilitators are included to inform about the experiences of implementing PanCareFollowUp Care (Figure 2). Items include, among others, drop-outs at different time-points, use of and experiences with the Survivorship Care Plan, and shared-decision making (Figure 1).”

3. Would also be helpful to briefly describe HEIQ questionnaire since this is the primary outcome (is it specific to cancer survivors or other populations, is there a cut-off for impairment, does it assess multiple domains)

Response: Thank you. We think it is a very valuable suggestion to describe the HEIQ questionnaire in the manuscript, as it is indeed our primary outcome.

The HEIQ is a well-validated and frequently used questionnaire to assess empowerment in patients. Originally, it has been developed for assessing empowerment in persons with chronic conditions. It is not specific for cancer patients and survivors but has been used successfully in cancer patients and survivors before (REF). The original HEIQ assesses eight different domains, of which the version previously used in cancer patients and survivors includes five: Social integration and support (5 items), Health service navigation (5 items), Constructive attitudes and approaches (5 items), Skill and technique acquisition (4 items) and Emotional distress (6 items). We chose to additionally include a sixth domain, Self-Monitoring and insight (6 items), since we considered it relevant for our study population. There is no cut-off for adequate or inadequate empowerment but a mean of the scale-items is calculated and indicates a higher or lower empowerment in the respective domain. In our study, we will compare before and after measurements of empowerment within participants, with the power calculation aimed at an effect size of 0.2.

We agree it would be helpful for readers to know more about the HEIQ questionnaire, and have made the following changes in our manuscript, including two additional references (page 11).

Previous version:

“The primary outcome for this study is empowerment measured by the Health Education Impact Questionnaire (HEIQ) (31). Empowerment has been defined by the EU Joint Action on Patient Safety and Quality of Care as a ‘multidimensional process that helps people gain control over their own lives and increase their capacity to act on issues that they themselves define as important’, a definition adapted from Lutrell et al. (32, 33). Empowerment has been selected as the primary outcome because childhood cancer survivors encounter several transition moments starting from diagnosis, after which a greater responsibility for their own health and care is required. It is essential that survivors receive the support they need to manage and advocate for their needs. Moreover, empowerment is important to manage future health problems.”

Current version (changes in orange):

“The primary outcome for this study is empowerment measured by the Health Education Impact Questionnaire (HEIQ) (31). Empowerment has been defined by the EU Joint Action on Patient Safety and Quality of Care as a ‘multidimensional process that helps people gain control over their own lives and increase their capacity to act on issues that they themselves define as important’, a definition adapted from Lutrell et al. (32, 33). Empowerment has been selected as the primary outcome because childhood cancer survivors encounter several transition moments starting from diagnosis, after which a greater responsibility for their own health and care is required. It is essential that survivors receive the support they need to manage and advocate for their needs. Moreover, empowerment is important to manage future health problems. We have included six of the eight scales of the HEIQ relevant to cancer survivors in our study (Social integration and support, Health service navigation, Constructive attitudes and approaches, Skill and technique acquisition, Emotional distress, Self-Monitoring and insight). The HEIQ has previously been used in cancer patient and survivor populations (34-36). It allows to calculate a mean for each scale indicating higher or lower empowerment in the respective domain within a participant compared to the baseline assessment.”

Additional references:

35. Brunet, J., et al., Measurement invariance of English and French Health Education Impact Questionnaire (heiQ) empowerment scales validated for cancer. Qual Life Res, 2015. 24(10): p. 2375-84.

36 .Osborne RH, Batterham R, Livingston J. The evaluation of chronic disease self-management support across settings: the international experience of the health education impact questionnaire quality monitoring system. *Nurs Clin North Am.* 2011;46(3):255-270, v.

4. Include in text specific measures that will be used to measure satisfaction and quality of life as secondary outcomes or reference Figure 1

Response: Thank you for noting that the specific measures for satisfaction, quality of life and other PREMs and PROMs were not connected to the corresponding part in the manuscript. Following the revised structure for the above-mentioned sections, we added a reference to Figure 1, which includes references to the specific measures used for each PREM or PROM (page 11).

Previous version:

“Secondary outcomes consist of a variety of patient-reported experiences and outcomes (PREMs and PROMs), such as satisfaction and quality of life.”

Current version:

“Secondary outcomes consist of a variety of patient-reported experiences and outcomes (PREMs and PROMs), such as satisfaction and quality of life (Figure 1).”

5. Add line in text describing how clinical outcomes will be identified (patient report, physician report, lab/imaging evals), and at what time points will they be assessed in addition to listing in Figure 1. Is it the provider at the time of the visit, patient, or someone else who reports these? And will any clinical outcomes they have ever had since cancer diagnosis be included at time-point 1 or just active clinical issues?

Response: Thank you for your important remark, we agree that we could improve this section by describing the clinical outcomes in more detail in the manuscript itself. As described in Figure 1, clinical outcomes will be collected by survivor self-report (through the Survivor Questionnaire) and by the healthcare provider (by the Treatment Summary, also including a section on past and current medical history, as well as through the medical history and physical exam performed at the clinic visit, and results from potential diagnostic tests). We will be able to take the source of these clinical outcomes into account when performing the analyses; outcomes reported by the survivor will be verified by the healthcare provider during the clinic visit and added to the Treatment Summary if accurate.

The Survivor Questionnaire and other parts of the PanCareFollowUp Care Intervention have previously been published with open access and can be found here: <https://pubmed.ncbi.nlm.nih.gov/34953441/>. Specifically, the Survivor Questionnaire consists of different sections, including current physical symptoms, current psychosocial symptoms and current medication use, but also past and current medical history and family history. Relevant items can be discussed at the clinic visit and added to the Treatment Summary, which also includes an overview of past and current medical history.

We have reflected the above in the manuscript through the following changes (page 12).

Previous version:

“Clinical outcomes are outcomes of symptoms and diseases and have been defined based on published or almost published guidelines of the IGHG and the PanCareFollowUp Recommendations. A total of 116 clinical outcomes were defined, which reflects the wide range of late effects that survivors may encounter affecting both physical health and psychosocial wellbeing (Figure 1). The number and range of pre-existing and newly detected health problems (symptomatic and

asymptomatic) per survivor will be described, including the results of clinical examinations (e.g. echocardiogram or blood tests).”

Current version (changes in orange):

“Clinical outcomes are outcomes of symptoms and diseases and have been defined based on published or almost published guidelines of the IGHG and the PanCareFollowUp Recommendations. A total of 116 clinical outcomes were defined, which reflects the wide range of late effects that survivors may encounter affecting both physical health and psychosocial wellbeing (Figure 1). Clinical outcomes include past and current medical history, are collected through survivor self-report in the Survivor Questionnaire (with verification at the clinic visit), and physician-report in the Treatment Summary, after the clinic visit and after potential diagnostic tests (Figure 2). The number and range of pre-existing and newly detected health problems (symptomatic and asymptomatic) per survivor will be described, including the results of clinical examinations (e.g. echocardiogram or blood tests).”

6. How will health economic outcomes be assessed? (participant survey, etc.)

Response: Thank you for this important question about one of our secondary outcomes. The health economic analyses, specifically the cost-effectiveness and cost-utility of the model, will be determined by comparing costs related to the PanCareFollowUp Care Intervention to benefits measured in terms of PREMs and PROMs. The health economic outcomes therefore focus on the costs experienced by the survivor as well as the healthcare provider/clinic, which are assessed in separate questionnaires.

We clarified this in the manuscript by slightly revising the text and referring to Figure 1 and 2 (page 12).

Previous version:

“The cost-effectiveness and cost-utility of the care model will be determined Health economic outcomes reflect the time, time off work and monetary investments made by the survivor, accompanying relatives or friends, the HCP and other staff in relation to the clinic visit while receiving or providing PanCareFollowUp Care. We do not take costs outside the clinic visit into account, i.e., costs related to possible (follow-up) primary care physician visits, mental health services, or referrals to other specialists outside the clinical setting. Costs related to the clinic visit, as associated with PanCareFollowUp Care, are compared to potential benefits measured in terms of PREMs and PROMs.”

Current version (changes in orange):

“The cost-effectiveness and cost-utility of the care model will be determined by using health economic outcomes (Figure 1). These reflect the time, time off work and monetary investments made by the survivor, accompanying relatives or friends, the HCP and other staff in relation to the clinic visit while receiving or providing PanCareFollowUp Care, and are collected using questionnaires (Figure 2). We do not take costs outside the clinic visit into account, i.e., costs related to possible (follow-up) primary care physician visits, mental health services, or referrals to other specialists outside the clinical setting. Costs related to the clinic visit, as associated with PanCareFollowUp Care, are compared to potential benefits measured in terms of PREMs and PROMs.”

7. Would be helpful to include description/definition of T1-T5 earlier in protocol before describing outcomes or make reference to figure earlier.

Response: Thanks for this suggestion, we agree that knowledge about the time points is relevant earlier on in the manuscript. This has been added to the manuscript as described below point 1.

8. *Would be helpful to add SQx questionnaire as appendix if this is a new measure specific to this protocol.*

Response: Thank you for this suggestion. As mentioned below point 5, the Survivor Questionnaire is available through open access. Each of the outcomes included is also described in Figure 1. Our suggestion is to mention this specifically in the part of the manuscript where the Survivor Questionnaire is mentioned as a data collection instrument (page 14).

Previous version:

“The main data collection instruments consist of the PanCareFollowUp Survivor Questionnaire (care), the Treatment Summary (care), medical history, physical examinations and diagnostic tests during and after the clinic visit (care), and additional online study questionnaires for survivors and HCPs (research).”

Current version (changes in orange):

“The main data collection instruments consist of the PanCareFollowUp Survivor Questionnaire (care), the Treatment Summary (care), medical history, physical examinations and diagnostic tests during and after the clinic visit (care), and additional online study questionnaires for survivors and HCPs (research). The Survivor Questionnaire and Treatment Summary are available through open access (29).”

9. *Typo page 14 line 58 should be "as well as" new diseases.*

Response: Thank you for pointing this out, we have corrected this mistake.

10. *Acronyms like SF-36, EQ-5D-5L are first used in statistics section but should be defined.*

Response: Thank you for mentioning this shortcoming, these acronyms are indeed only mentioned in the legend of Figure 1. We have adapted the manuscript accordingly for the SF-36 and the ICECAP-A at the first point they are mentioned in the manuscript (page 15). Importantly, the EQ-5D-5L is not an abbreviation and is the correct term to use when referring to the instrument in general (as described in the article: <https://pubmed.ncbi.nlm.nih.gov/32620995/>).

Previous version:

“The estimated benefits of the intervention in terms of empowerment (HEIQ), quality of life (SF-36, EQ-5D-5L, ICECAP-A), and other outcomes are compared to the additional costs of implementing the PanCareFollowUp Care Intervention.”

Current version (changes in orange):

“The estimated benefits of the intervention in terms of empowerment (HEIQ), quality of life (Short-Form 36 (SF-36), EQ-5D-5L, ICEpop CAPability measure for Adults (ICECAP-A)), and other outcomes are compared to the additional costs of implementing the PanCareFollowUp Care Intervention.”

VERSION 2 – REVIEW

REVIEWER	Horton, Susan University of Waterloo, School of Public Health and Health Systems
REVIEW RETURNED	29-Sep-2022

GENERAL COMMENTS	Thank you for the clarifications regarding cost-effectiveness, I now understand more clearly what is being proposed. If the cost-effectiveness component is maintained (and see my further concerns below) I believe it would be appropriate to clarify question 4) to read "what is the cost-effectiveness of follow-up after 6 months, and the projected cost-effectiveness modelled over a longer period". p17: What is the comparator, for calculation of the ICER? I did not see a control group mentioned (did I miss this?). How is it possible to calculate an ICER without this? One can do a cost analysis perhaps (is the intervention likely to increase or decrease costs over the six month period - but even that would require data on patients in the absence of an intervention). It looks as if the authors aim to find the cost per unit change in empowerment of patients. The authors mention non-uniformity of thresholds against which to compare the ICER as a concern. However that is the least of my worries. I am not aware of other studies examining the cost per unit increase in empowerment as the primary outcome; if there are such, it would be important to cite them so as to know whether this intervention is good value for money. If instead the authors plan to link empowerment to a more direct measure of utility, it would be helpful to spell that out, and to provide references linking empowerment to such a measure. I did a quick search to look for studies of cost-effectiveness of improving empowerment of patients. I ran across one for an intervention which aims to increase empowerment in cardiac patients (Yu et al, JAMA Network Open April 5, 2022), which did a 2-year RCT including measurement of health outcomes. But that is a very different study design to the present one. In summary, perhaps I am still missing some key information? I still am very sceptical of the value and validity of the proposed cost-effectiveness analysis. This does not detract from other components of the study which may be very useful (I leave that to the other reviewer for comment).
--

REVIEWER	Rodwin, Rozalyn Yale School of Medicine
REVIEW RETURNED	14-Oct-2022

GENERAL COMMENTS	The authors have adequately addressed and clarified all questions. I have no further suggested revisions.
---

VERSION 2 – AUTHOR RESPONSE

Comments by reviewer 1

1. Thank you for the clarifications regarding cost-effectiveness, I now understand more clearly what is being proposed. If the cost-effectiveness component is maintained (and see my further concerns

below) I believe it would be appropriate to clarify question 4) to read "what is the cost-effectiveness of follow-up after 6 months, and the projected cost-effectiveness modelled over a longer period".

Response: We thank the reviewer for this excellent suggestion. We agree with the refined wording of question 4 to clarify the time horizon of the health economic analyses planned, a) within the data collected in the project, and b) beyond based on a health economic modeling approach. We have adjusted the text accordingly, see pages 8 and 12 of the revised manuscript.

Previous version on pages 8 and 12:

“4) What is cost-effectiveness and cost-utility of implementing PanCareFollowUp Care relative to usual care from the perspective of survivors, health care providers (HCPs), and society at large?”

Current version on pages 8 and 12 (changes in orange):

“4) What are the short-term (six months) and projected long-term costs per unit change of empowerment and other outcomes after cost-effectiveness and cost-utility of implementing PanCareFollowUp Care relative to usual care from the perspective of survivors and health care providers (HCPs) society at large?”

2. p17: What is the comparator, for calculation of the ICER? I did not see a control group mentioned (did I miss this?). How is it possible to calculate an ICER without this? One can do a cost analysis perhaps (is the intervention likely to increase or decrease costs over the six month period - but even that would require data on patients in the absence of an intervention).

Response: Thank you for bringing up this new point. We see that our proposal for the analysis of health economic outcomes is not entirely clear. We will base our analyses of the costs and benefits (in terms of patient-reported experiences and outcomes for survivors, including empowerment) on within-subject changes until six months of follow-up, and on model-based evaluations for longer-term predictions. We have reflected this in the manuscript by adding the following section on page 15 and modifying sections on page 1 and 8.

Addition on page 15:

“The analysis of costs and benefits will be based on within-subject changes until six months of follow-up, and on model-based evaluations for longer-term predictions.”

Previous version on page 1:

Evaluating the feasibility and cost-effectiveness of implementing person-centred follow-up care for childhood cancer survivors in four European countries: the PanCareFollowUp Care prospective cohort study protocol

Current version on page 1 (changes in orange):

Evaluating the feasibility, and cost-effectiveness and costs of implementing person-centred follow-up care for childhood cancer survivors in four European countries: the PanCareFollowUp Care prospective cohort study protocol

Previous version on page 8:

“The overall aim of the PanCareFollowUp Care Study is to evaluate the feasibility and (cost-)effectiveness of implementing PanCareFollowUp Care as usual care for adult survivors of childhood cancer in four study sites in four European countries.”

Current version on page 8 (changes in orange):

“The overall aim of the PanCareFollowUp Care Study is to evaluate the feasibility, and (cost-)effectiveness and costs of implementing PanCareFollowUp Care as usual care for adult survivors of childhood cancer in four study sites in four European countries.”

3. It looks as if the authors aim to find the cost per unit change in empowerment of patients. The authors mention non-uniformity of thresholds against which to compare the ICER as a concern. However that is the least of my worries. I am not aware of other studies examining the cost per unit increase in empowerment as the primary outcome; if there are such, it would be important to cite them so as to know whether this intervention is good value for money. If instead the authors plan to link empowerment to a more direct measure of utility, it would be helpful to spell that out, and to provide references linking empowerment to such a measure. I did a quick search to look for studies of cost-effectiveness of improving empowerment of patients. I ran across one for an intervention which aims to increase empowerment in cardiac patients (Yu et al, JAMA Network Open April 5, 2022), which did a 2-year RCT including measurement of health outcomes. But that is a very different study design to the present one.

Response: Indeed, we would like to estimate the costs per unit change in empowerment of patients before and after the intervention. We agree that empowerment is not a common outcome that has been used before, but we deem it highly relevant in the present context of follow-up care and consider it a largely understudied outcome especially for cancer survivors. Survivors of childhood cancer highlighted this during the project application process, and it has also been underlined by the European Patients' Forum (see: https://www.eu-patient.eu/globalassets/campaign-patient-empowerment/epf_briefing_patientempowerment_2015.pdf).

We agree that it will be difficult to analyze the cost-effectiveness of PanCareFollowUp Care in our study, because we miss a control arm. The dilemma in survivorship care is that no RCTs have been performed before its implementation. Because of existing, internationally endorsed guidelines for the need of early surveillance of health problems, it is at this point not ethical to perform randomized trials withholding surveillance anymore. For cancer survivorship care, multiple studies estimated that surveillance could be cost-effective to detect important clinical outcomes (Furzer et al., 2019; Gini et

al., 2019; Yeh et al., 2022; Ykema et al., 2022; see references at the end of this letter). Even in the reality of a lack of survivorship care for many survivors worldwide, this is not due to discussion about its importance, but do to other implementation-related factors (e.g., financial, logistic, political) (Essig et al., 2012). This is the reason that we chose for an observational study design where we would like to calculate the costs and analyze the benefits using a before-after assessment of the main outcome empowerment.

In conclusion, we agree that an official cost-effectiveness analysis in our current study design could be too ambitious. However, for future implementation of follow-up care, an overview of the costs and potential benefits could be essential. Therefore, we propose to follow the reviewer's suggestion and perform a cost evaluation instead of a cost-effectiveness analysis. We have reflected this in the manuscript by making the changes below.

Previous version on page 12:

"The cost-effectiveness and cost-utility of the care model will be determined by using health economic outcomes (Figure 1)."

Current version on page 12 (changes in orange):

"The costs associated with implementing -effectiveness and cost-utility of the care model will be determined by using health economic outcomes (Figure 1)."

Previous version on page 15 and 16:

"For the health economic evaluation, we will calculate incremental cost-effectiveness ratios (ICERs) for different outcomes. The estimated benefits of the intervention in terms of empowerment (HEIQ), quality of life (Short-Form 36 (SF-36), EQ-5D-5L, ICEpop CAPability measure for Adults (ICECAP-A)), and other outcomes are compared to the additional costs of implementing the PanCareFollowUp Care Intervention. Costs include resources incurred at the level of the hospital and the survivor. At the hospital level, we measure the time of physicians and other hospital staff for tasks related to the clinic visit and the follow-up call, costs for diagnostic and screening tests and other consumables for the clinic visit. At the survivor level, we measure the time investment and travel costs of survivors and relatives or friends, and loss of productive time at the workplace or in education. These costs are investigated separately on each level, hospital and survivor, as well as on an aggregated level. To account for statistical uncertainty in the cost data, we will apply a bootstrap approach using empirical and/or theoretical distributions on different cost positions. Results are displayed in a cost-effectiveness plane. Since there are no uniform ceiling values on ICERs across countries (and for the different outcomes), we will also show cost-effectiveness acceptability curves, which account for statistical uncertainty in the ICERs and in the ceiling values.

The calculation of ICERs needs to be interpreted in light of the relatively short follow-up period of six months within the study. This implies that the cost-effectiveness analysis mainly focuses on short-run effects, while longer-run effects of PanCareFollowUp Care on outcomes such as survival cannot be measured within the study. Moreover, effects on other outcomes such as quality of life may be small. In order to provide information about the potential medium- to long-run effects, we will complement

our analysis with a model-based economic evaluation approach using data from this study as well as information from the literature on longer-term effects of follow-up interventions and patient pathways, as well as related cost estimations. This will allow us to gain a more comprehensive picture on the cost-effectiveness of PanCareFollowUp Care.”

Current version on page 15 and 16 (changes in orange):

“For the health economic evaluation, we will calculate incremental cost-effectiveness ratios (ICERs) the costs associated with the implementation of the PanCareFollowUp Care Interventions in order to achieve change in for different outcomes. The estimated benefits of the intervention are measured in terms of empowerment (HEIQ) and quality of life (Short-Form 36 (SF-36), EQ-5D-5L, ICEpop CAPability measure for Adults (ICECAP-A)), compared to the additional costs of implementing the PanCareFollowUp Care Intervention.

Costs include resources incurred at the level of the hospital and the survivor. At the hospital level, we measure the time of physicians and other hospital staff for tasks related to the clinic visit and the follow-up call, costs for diagnostic and screening tests and other consumables for the clinic visit. At the survivor level, we measure the time investment and travel costs of survivors and relatives or friends, and loss of productive time at the workplace or in education. These costs are investigated separately on each level, hospital and survivor, as well as on an aggregated level. To account for statistical uncertainty in the cost data, we will apply a bootstrap approach using empirical and/or theoretical distributions on different cost positions. Results are displayed in a cost-effectiveness plane. Since there are no uniform ceiling values on ICERs across countries (and for the different outcomes), we will also show cost-effectiveness acceptability curves, which account for statistical uncertainty in the ICERs and in the ceiling values.

The calculation of cost per unit change of outcomes ICERs needs to be interpreted in light of the relatively short follow-up period of six months within the study. This implies that the cost evaluation effectiveness analysis mainly focuses on short-run effects, while longer-run effects of PanCareFollowUp Care on outcomes such as survival cannot be measured within the study. Moreover, effects on other outcomes such as quality of life may be small. In order to provide information about the potential medium- to long-run effects, we will complement our analysis with a model-based economic evaluation approach using data from this study as well as information from the literature on longer-term effects of follow-up interventions and patient pathways, as well as related cost estimations. This will allow us to gain a more comprehensive picture on the costs associated with the implementation-effectiveness of PanCareFollowUp Care.”

4. In summary, perhaps I am still missing some key information? I still am very sceptical of the value and validity of the proposed cost-effectiveness analysis. This does not detract from other components of the study which may be very useful (I leave that to the other reviewer for comment).

Response: We can understand that our ambitions regarding the financial consequences of implementation of follow-up care were perhaps too high. We are thankful for the critical review of our study protocol manuscript and hope that our abovementioned responses and modifications may

convince the reviewer of the value of the proposed analysis, within the possibilities available for data collection and analysis within this European study.

Comments by reviewer 2

The authors have adequately addressed and clarified all questions. I have no further suggested revisions.

Response: We are very grateful for your questions and suggestions, and are confident that they helped improve the quality of our manuscript. Thank you for your review.

References

Essig et al. Follow-up programs for childhood cancer survivors in Europe: a questionnaire survey. PLoS One. 2012;7(12):e53201.

Furzer et al., Cost-Utility of Early Breast Cancer Surveillance in Survivors of Thoracic Radiation-Treated Adolescent Hodgkin Lymphoma. J Natl Cancer Inst. 2020;112(1):63-70.

Gini et al., Cost-Effectiveness of Colonoscopy-Based Colorectal Cancer Screening in Childhood Cancer Survivors. J Natl Cancer Inst. 2019;111(11):1161-1169.

Yeh et al., Breast Cancer Screening Among Childhood Cancer Survivors Treated Without Chest Radiation: Clinical Benefits and Cost-Effectiveness. J Natl Cancer Inst. 2022;114(2):235-244. doi: 10.1093/jnci/djab149.

Ykema et al., Cost-effectiveness of colorectal cancer surveillance in Hodgkin lymphoma survivors treated with procarbazine and/or infradiaphragmatic radiotherapy. Cancer Epidemiol Biomarkers Prev. 2022;EPI-22-0019.

VERSION 3 – REVIEW

REVIEWER	Horton, Susan University of Waterloo, School of Public Health and Health Systems
REVIEW RETURNED	31-Oct-2022
GENERAL COMMENTS	I think this will be an interesting and useful study, and am happy that the authors have updated the methods.